# Novel Approach to Synthesis of AgZnS and TiO_2_ Decorated on Reduced Graphene Oxide Ternary Nanocomposite for Hydrogen Evolution Effect of Enhanced Synergetic Factors

**DOI:** 10.3390/nano12203639

**Published:** 2022-10-17

**Authors:** Jingjing Zhao, Md Nazmodduha Rafat, Chang-Min Yoon, Won-Chun Oh

**Affiliations:** 1College of Pharmaceutical Sciences, North China University of Science and Technology, Tangshan 063210, China; 2Department of Advanced Materials Science & Engineering, Hanseo University, Seosan-si 31962, Korea; 3Department of Chemical and Biological Engineering, Hanbat National University, Daejeon 34158, Korea

**Keywords:** ternary nanocomposite, AgZnS-G-T, synergetic factor, ultrasonication, hydrogen evolution

## Abstract

In this work, a novel ternary nanocomposites AgZnS-TiO_2_-reduced graphene oxide (RGO) was successfully synthesized by a facile soft ultrasonic-reduction condition as low as 70 °C. During the ultrasound reaction, the reduction of GO and the growth of AgZnS and TiO_2_ crystals occurred simultaneously in conjunction with the deposition of AgZnS and TiO_2_ crystals onto the surface of the graphene. The synthesized nanocatalysts were characterized by XRD, SEM, TEM, EDX, Raman spectroscopy, XPS, UV–Vis DRS, photoluminescence spectrometer, and photocurrent and CV. The AgZnS-G-T was shown as catalytic HER with some synnegetic factors such as pH-universal, temperature, and ultrasonic condition. After 4 h, it was observed that AgZnS-TiO_2_-RGO has the highest efficiency of photocatalytic activity through hydrogen production by water splitting, which achieved the highest hydrogen evolution rate of 930.45 μmol/g at buffer solution (pH = 5), which was superior to AgZnS-G (790.1 µmole/g) and AgZnS (701.2 µmole/g). Such a significant hydrogen evolution amount far exceeded that of undoped TiO_2_ and RGO. The H_2_ evolution amounts increased significantly at ultrasonic irradiation power of 80 MHz. AgZnS-G-T demonstrates the higher H_2_ evolution amounts of 985 µmole/g at 80 MHz. Its photocatalytic hydrogen-evolution activity remained at a high level over four cycles (16 h) nanoparticle.

## 1. Introduction

The exhaustion of the global supply of fossil fuels has prompted numerous schools to hunt for other renewable energy sources. Hydrogen has been encouraged as an attractive alternative for overcoming the current environmental problems and energy crisis, owing to its high energy content, zero carbon emissions, and ease of storage and recyclability. Titanium dioxide is an important n-type semiconductor with a band-gap of 3.2 eV extensively used in photocatalytic degradation of organic pollutants, hydrogen production, and solar cells. However, its wide band-gap, rapid recombination of the photo-generated electron (eCB^−^) and hole (hVB^+^) pairs, low quantum yield, and a low response to visible light enormously limit its practical applications. A lot of scholars have focused on improving the efficiency of photocatalytic H_2_ production utilizing TiO_2_ as the photocatalysts [1,2]. However, it is difficult for a single semiconductor to achieve good catalytic performance. Doping, metal particle deposition, or their combination with other materials into TiO_2_ could produce highly photoactive compounds [2].

Metal sulfides, a significant class of narrow-bandgap semiconductor materials, are excellent cocatalysts widely used to modify wide-bandgap semiconductor photocatalysts. Transition metals incorporated with sulfides such as CdS [3,4], PbS [5,6], Ag_2_S [7,8], NiS_2_ [9,10], and WS_2_ [11,12] are considered excellent candidates of promising approaches for hydrogen evolution due to the highly efficient catalysts. Recent research has shown that ternary metal sulfides (TMS) are a promising alternative for various electrocatalytic reactions due to their high electronic conductivity and excellent redox reversibility [13]. Compared with monometallic sulfides, TMSs possess more attractive structural advantages as electrocatalysts. Several recent studies have verified that TMS-based electrocatalysts have great potential as photoelectrocatalysts for water-splitting [13,14,15]. For instance, nickel, cobalt, and iron metals doped into the crystalline MoS_2_ induce hydrogen evolution reaction (HER) activity enhancement. Compared to the catalytic activities of the MMoS_x_ and MoS_2_ catalysts under the same conditions, the MMoS_x_ catalysts turned out to be the more active catalysts [14]. Dai et al. proved that the HER activity of the M–MoS_2_ (M = Fe, Co, and Ni) catalysts is higher than pristine MoS_2_ [13]. The electrocatalytic HER activity of Cu_2_MoS_4_, as a ternary sulfide, was also first reported by Tran et al. [15]. Mixed metal sulfides with other compositions have also been reported, such as ZnCoS [16], Ag_2_WS_4_ [17], CoNiWS [18], and ZnIn_2_S_4_ [19,20], showing interesting electrochemical performance for electrocatalysis. Doping with a second metal is thought to play a significant role in optimizing the free energy of hydrogen adsorption, resulting in higher HER activity than undoped monometallic sulfides [21]. The first-principles calculations based on density functional theory provide a possible understanding of surface interactions between RGO decoration with other transition metal [22,23].

Synthesis methods of TMS are often present in the literature but require a long reaction time and high pressure. For instance, Staszak-Jirkovský reported that CoMoS_x_ chalcogenides were synthesized by the sol–gel method in a few days [24,25]. Wang stated that a hydrothermal process synthesized NiMoS_4_ at 160 °C for 9 h [25]. However, a facile and general method for controlling TMS synthesis that lowers the reaction temperature simplifies the synthesis procedure and shortens the reaction time has not been realized. In this work, AgZnS ternary metal sulfide is synthesized by a simple ultrasonication approach.

The catalytic activities of nanometer materials are determined by an appropriate synthesis route, because their features and properties are decided by their structure. However, no reports regarded the synthesis of graphene-AgZnS-TiO_2_ nanocomposite by an ultrasonication method for the enhanced photocatalytic effects. We aimed at contributing a new synthesis approach to the limited synthesis methods. When solutions are subjected to ultrasound irradiation unsheltered, adequate acoustic energy can drive the generation of novel nanostructures to occur. This nanocomposite has more benefits than the previous nanoparticles that were prepared by simple mechanically mixing [26]. The ultrasonic approach produces nanoparticles with considerably narrower size distribution and a larger surface area, both of which contribute to the enhanced catalytic activity toward HER. Furthermore, adding ultrasound waves increases the hydrogen evolution amount during the illumination.

## 2. Experimental Section

### 2.1. Materials

All chemicals used were commercially available. The standard-grade chemicals were used without further purification to modify this material. We purchased Silver nitrate (AgNO_3_, 98.5%) from Samchun Pure Chemical Co., Ltd. (Pyungtaek, Korea), Titanium (IV) Butoxide [Ti(OCH_2_CH_2_CH_2_CH_3_)_4_, 97.0%] from Daejung Chemicals & Metals Co., Ltd. (Shiheung, Korea), Commercial grade sodium sulfide pentahydrate (Na_2_S·5H_2_O), 98.0%, and zinc chloride (ZnCl_2_), 98.0%, from Yakuri Pure Chemical Co., Ltd. (Osaka, Japan), and Duksan Pure Chemical Co., Ltd. (Ansan-si, Korea).

### 2.2. Preparation Process for Different Photocatalysts

We used the ultrasonic preparation method to synthesize the AgZnS-G-TiO_2_ photocatalyst. A three-step combined method was followed in this preparation method. First, AgZnS is required. A total of 1.74 mg of silver nitrate was dissolved in 50 mL water during preparation. We took 0.82 mg zinc chloride (ZnCl_2_) contained in 50 mL water and prepared it by stirring for 30 min under 30 °C; then, 1.01 mg sodium sulfide pentahydrate (Na_2_S·5H_2_O) was dissolved in 50 mL water. After that, zinc chloride and sodium sulfide pentahydrate solutions were mixed dropwise (2 drops/s) in the previously made silver nitrate solution. Then, the mixture was sealed with aluminum fuel and stirred overnight at 80 °C. Finally, the ultrasonic process was used for 5 h with 60 Hz amplitude and 90 °C temperature. Then, the solution was washed several times with DI water and dried at 100 °C for 12 h, followed by calcination at 600 °C. This collected sample was named AgZnS.

Next, we followed the same process to synthesize AgZnS-G. Graphene was produced by the Hummer method. We dissolved 1.0 g graphene in 50 mL of water and 30 mL ethanol, 94.0%, and then used the ultrasonic process for 2 h. Graphite oxide was reduced to graphene by ultrasonic treatment. After that, this homogeneous solution was combined with the previously made AgZnS solution and stirred for 1 h at 100 °C. Then, the solution was sonicated for 5 h as in the previous condition, washed several times with DI water, and dried at 110 °C for 12 h, followed by calcination at 600 °C for 3 h. The collected sample was named AgZnS-G.

The third step to be considered is the addition of TiO_2_. At first, 1.8 mL of Titanium (IV) Butoxide, 97.0%, were treated with 4 mL of acetic acid, 97.0%, and 30 mL ethanol, 94.0%. We prepared the AgZnS-G solution by following the same process. Then, the TiO_2_ solution was added dropwise into the AgZnS-G solution and stirred for 1 h. Sonication was required one more time for 3 h with the same condition. After that, the solution was washed several times and dried at 120 °C for 12 h. Finally, we calcinated the sample at 600 ℃ for 3 h. The collected sample is AgZnS-G-T.

### 2.3. Instrumental Characterization

To figure out the crystalline structure, we analyzed X-ray diffraction (XRD) data using an X-ray diffractometer (XRD-6000, SHIMADZU, Kyoto, Japan) equipped with a Cu Kα X-ray source (1.5406 Å). The morphology and structure of the nanomaterial were analyzed by SEM images (JSM-5600 JEOL, Tokyo, Japan). Using an Energy Dispersive X-ray (EDX) incorporated with SEM, we identified the chemical compositions and bonding. TEM images analyzed the morphology of each sample. We used a transmission electron microscope (TEM, Hitachi HT7700, operated at 100 kV, Tokyo, Japan) to analyze the TEM images. We carried out X-ray photoelectron spectroscopy (XPS) measurements on a KRATOS AXIS SUPRA with a monochromatized Al Kα X-ray source working at 10 kV, 1500 W, and pass energy of 40 eV. Raman spectroscopy was operated by a Confocal Raman imaging system with a 633 nm laser for excitation (Renishaw in Via Reflex). We analyzed UV–Vis diffuse reflectance spectra (DRS) with an ultraviolet-visible (UV–Vis) spectrophotometer (SHIMADZU UV–2600) at a wavelength of 200 to 800 nm. We calculated band-gap energies of photocatalysts by applying a modified Kubelka–Munk function from UV–Vis DRS data. Electrochemical impedance spectroscopy (EIS) in the frequency range of 100 kHz to 0.01 HZ and an AC amplitude is 10 mV. We measured photoluminescence (PL) spectra using an Edinburgh Instruments FLS920P equipped with a Xe-lamp-920 at room temperature under the excitation of 360 nm. The electrical performance of the photocatalysts was analyzed by cyclic voltammetry test (VersaStat4-400) with a standard three-electrode system.

### 2.4. Photocatalytic Measurement

To carry out the photocatalytic experiment under ambient conditions, we dissolved 0.1 g of synthesized photocatalyst in 120 mL DI water. A handmade visible-light source derived from an 8-watt lamp (Fawoo, Lumidas-H, Bucheon, Korea, λ ≥ 420 nm) with a filter (Kenko Zeta, transmittance > 90%). It was necessary to carry out experiment for photocatalytic hydrogen production under visible light for 4 h in a glass reactor to prevent radiation below about 410 nm. A self-made equipment and a detector (hydrogen detector: Minimax (X13010683) XP H_2_ sensor) were applied to measure the photocatalytic hydrogen production. We also carried out this experiment with pH and temperature control. Cavity implosion caused localized temperatures and pressures (inside the bubble) to reach as high as 5000 K and 1000 atm, respectively. It is generally believed that these severe circumstances result in the formation of highly reactive species such as hydroxyl (OH·) and hydrogen (H·) radicals.

### 2.5. Electrical Performance Test

Electrochemical measurement has been analyzed by a cyclic voltammeter (CV) analyzer (VersaStat4-400) workstation with a conventional three-electrode system. The cyclic voltammetry (CV) test was carried out at the scan rate of 100 mV s^−1^. The working electrode was the photocatalysts modified FTO glass. We dissolved 0.1 g of sample in 10 mL ethanol and 0.05 mg of ethylcellulose as binding material. Then, the paste catalyst was coated onto the surface of the FTO glass (1 cm^2^). Finally, the FTO working electrode covered with sample was obtained by drying at room temperature. As a counter electrode, we used a Pt. and Ag/AgCl electrode as a reference electrode. We used water as an electrolyte. Additionally, to investigate the presence of electron-hole pair generation in the prepared nanocatalyst, photocurrent studies were employed by PGP201 Potentiostatic A41A009. Photocurrent responses of the nanophotocatalyst as light on and off were tested at open-circuit potential, with simulated solar light illumination provided by a 150 W Halogen bulb. The photocurrent value was tested using an SLS301-stabilized benchtop Tungsten–Halogen light source with a 150W Halogen bulb. The wavelength range is 360–2700 nm. Output optical power is >1.6 W. Linear sweep voltammetry (LSV) was performed in 1 M KOH. Nitrogen purged prior the experiment. The linear sweep voltammetry (LSV) tests in the potential ranging from open-circuit voltage to −1 V versus Hg/HgO were conducted at a scan rate of 0.01 v/s. The potentials converted to reversible hydrogen electrode (RHE) according to the equation: E(RHE) = E_(Hg/HgO__)_ + 0.059pH + E^0^(Hg/HgO). The preparation procedure and electrochemical experimental method of the samples are presented in Figure 1.

## 3. Results and Discussion

### 3.1. Characterizations of Photocatalysts

Figure 1 presents the XRD patterns of the AgZnS, AgZnS-G, and AgZnS-G-T composites. The characteristic peaks of TiO_2_ can be found at 2θ = 25.4, 44.4, 77.6, and 81.7° corresponding to (101), (200), (215), and (224) crystal planes reflections that correspond to the anatase crystal phase (JCPDS PDF#: 21-1272). Nine characteristic peaks of AgZnS can be found at 2θ = 26.7, 30.9, 38.1, 44.3, 55.1, 64.6, 73.4, 77.5, and 81.8° corresponding to the (100), (101), (110), (002), (111), (200), (211), (103), and (112) crystal faces reflections that correspond to the anatase crystal phase (JCPDS PDF#: 80-0020). No single phase of ZnS was found from the XRD pattern. The study of the combined complex of ZnS can be seen from our previously published work [27]. The synthesized composite compound was elementally analyzed using DEX. The results for these are presented in Figure 1b,c. These findings suggest that ultrasonic synthesis is an effective method to synthesize AgZnS nanoparticles. Due to their weakness, there were hardly any diffraction peaks of RGO to be found relative to the strong diffraction pattern of TiO_2_ and AgZnS, which was possibly because of the low crystallinity and content of RGO in the prepared samples [28]. Raman and TEM data from subsequent research could establish the presence of GO in the composites.

The pure AgZnS particles exhibit an agglomerated state. Pure graphene is likely to restack or agglomerate easily due to its high cohesive energy, van der Waals force interactions and strong stacking tendency. Figure 2a–f depicts the plane surface of AgZnS and TiO_2_ that were homogeneously attached together. The shapes of TiO_2_ and AgZnS particles are mostly irregularly spherical types. The AgZnS has a significantly greater volume than TiO_2_. After ultrasonic treatment, the overall morphology of the AgZnS-TiO_2_-RGO sample demonstrates that the AgZnS and TiO_2_ particles are uniformly anchored on RGO sheets. Ultrasonic coupled behavior is a helpful factor for the uniform distribution of microscale particles on the graphene surface. Under ultrasonic conditions, TiO_2_ and AgZnS microspheres can be well-dispersed and uniformly attached to the surface of the plate-like RGO sheets, which is beneficial for interfacial electron transfer. Successful interfacial contact between graphene, AgZnS, and TiO_2_ led to efficacious catalytic activity for the H_2_O reduction to H_2_. Differences in crystalline size for samples AgZnS, AgZnS-G, and AgZnS-G-T could also be observed in SEM images of Figure 2a–c. Flake-like RGO is almost transparent. RGO has an asymmetrical structure that is fragmented into different orders. AgZnS and TiO_2_ irregularly covered the RGO surface, indicating that the RGO nanosheets arrange for a good raised area for the nucleation and subsequent development of AgZnS and TiO_2_ nanoparticles. Graphene can prevent AgZnS and TiO_2_ accumulation; AgZnS and TiO_2_ also prevent graphene from accumulating conversely. The TEM images of Figure 2d–f show that RGO is translucent and the AgZnS or TiO_2_ nanoparticles were irregular dark imaged compounds. The average size of the prepared nanophotocatalyst is in the range of 20 to 40 nm. Moreover, the mass ratios of C, O, S, Ti, Zn, and Ag in AgZnS-G-T are validated by elemental diffraction analysis (EDS) and are presented in Figure 1b,c. These EDS results provide further support for the presence of C, O, S, Ti, Zn, and Ag in the nanocomposite.

Figure 3a shows the full range of XPS spectra corresponding to Ag 3d at (365–376) eV, Zn 2p at (1019–1025) eV, S 2p at (161–166) eV, C 1s at (282–288) eV, O 1s at (528–535) eV, and Ti 2p at (456–466) eV, confirming the presence of the AgZnS and TiO_2_ loading on GO. The peaks of O1s consisted of four peaks mainly occurring at 530.52 eV of O-Me, 532.57 eV of O=C, 533.96 eV of O–C=O, and 540.51 eV of O–OH (Figure 3b). The binding energy values for O binding were almost identical to the results of the previous study [29]. The binding energy of S was at the 163.53 eV binding energy region, which corresponds to the S 2p_1/2_ and S 2p_3/2_ spectra shown in Figure 3c. Ti^4+^ in TiO_2_ had two peaks at 459.27 and 464.88 eV, which are related to Ti 2p_3/2_ and Ti2p_1/2_ as shown in Figure 3d. The binding energy for Zn was calculated from the high-intensity peaks at the 1022.53 and 1045.61 eV positions, which are from the Zn 2p_3/2_ and Zn 2p_1/2_ bonding orbital, as shown in Figure 3e. The peaks located at (368.17 and 374.15) eV in Figure 3f are ascribed to Ag 3d. It is characteristic peak of zero-valence Ag. The binding energy peaks of C 1s at 283.75 and 284.43 eV are ascribed to the non-oxygenated C (C–C and C=C) in aromatic rings and the epoxy group of C–O–C. The peak of oxygenated carbon in RGO decreased significantly even disappeared, indicating that GO was gradually reduced to RGO in the preparation process. The O 1s spectra located at (530.5 and 532.8) eV are ascribed to the C=O, C–O–C, and O–C–OH. The binding-energy value of Ti 2p_3/2_ is 465.0 eV, and Ti 2p_1/2_ is 459.2 eV, which matches well with the titanium (IV) species [29,30]. The XPS spectra of the C 1s de-convoluted into five peaks, with C–C at 284.24 eV, C=C at 284.81 eV, C–O at 285.64 eV, C=O at 289.14 eV, and O–C=O group at the 296.01 eV regions for graphene [31]. The peaks of O1s consisted of four peaks mainly occurring at 530.03 eV of O–C, 530.57 eV of O–Me, 531.79 eV of O=C, and 532.93 eV of O–C=O [31].

Figure 4b represents the Raman spectroscopy of AgZnS, AgZnS-G, and AgZnS-G-T. Raman peaks corresponding to the AgZnS can be observed at 1343 and 1580 cm^−1^. There is a considerable overlapping peak between AgZnS and RGO. A typical Raman spectrum of graphene exhibits two main bands: the D (1351 cm^−1^) and G (1601 cm^−1^). The D-band peak (A1g-symmetry) reflects irregular elongation of the sp^2^, while the G-band peak (E2g-symmetry) can reflect the appearance of the first-order dispersion of C–C resonance of sp^2^. This further verifies the presence of RGO. TiO_2_ at 390, 500, and 620 cm^−1^, which theoretically represents the appearance of the Eg, B1g, and A1g modes of TiO_2_ [31]. The Eg peak is assigned to the symmetric stretching vibration, and the B1g and A1g peaks are related to the symmetric bending vibration of the O–Ti–O.

The band-gap of the nanocomposites was analyzed by UV–visible DRS spectrometry. The UV–vis spectra of the prepared samples are displayed in Figure 4a. The band-gap energies of these prepared samples can be estimated by Equation (1):(1)ahv=A(hv−Eg)12
where *α*, *h*, *ν*, and *E_g_* are the absorption coefficient, Planck constant, light frequency, and band-gap, respectively [31]. The band gaps (*E_g_*), which were estimated from the extrapolated values of the tangents on the wavelength axis to the plots of (*αhν*)½ versus *hν* Figure 4a, were 1.60, 2.25, and 2.40 eV for AgZnS-G-T, AgZnS-G, and AgZnS, respectively. According to the Nernst equation, it corresponds to ΔE° = 1.23 V per transferred electron. Theoretically, a semiconductor with band gap energy (Eg) larger than 1.23 eV can generate electrons and holes trigger the hydrogen evolution reaction (HER) and oxygen evolution reaction (OER) under irradiation. ZnS is a large band gap semiconductor. TMS features more metal centers and, as a result, greater tunable band gaps. Ag doped ZnS declined the band gap values. The decline in the optical band gap values with doping Ag might have been due to sp-s/sp-d exchange interactions among the band electrons of the “s “or “d” orbitals and localized electrons in the ZnS host lattice. Coupling with RGO and TiO_2_ has lower band gap energy than in pure AgZnS. A redshift of the band gap transition is observed. Narrower band-gap energy leads to a much broader absorption in the visible light region, making the AgZnS-G-T composites more abundant in photoinduced carriers. In addition, narrower band-gap energy can energize to generate more electron-hole pairs under the similar visible-light illumination, which could result in higher photocatalytic activity and has great potential for photocatalytic water splitting [32,33].

Photoluminescence (PL) spectra are used to investigate the efficiency of photoexcited charge carrier trapping, immigration, and transfer of semiconductors, therefore revealing the annihilation of the photo-generated electron (eCB^−^)–hole (hVB^+^) pairs in semiconductors. The emission peak of pristine AgZnS centered at 714.2 nm. However, the peaks of AgZnS-G and AgZnS-G-T are hardly seen. Figure 4c shows the intensity of the PL spectra of AgZnS-G and AgZnS-G-T. Nanocomposite decreases compared to that of pure AgZnS, indicating that incorporating TiO_2_ and RGO leads to a decrease in the recombination rate of electrons and holes in the nanocomposite under light irradiation. This phenomenon is attributed to the fact that the photoexcited electrons are transferred from the conduction band of TiO_2_ to AgZnS and then transferred to RGO sheets due to its excellent electronic conductivity, preventing direct recombination of electrons and holes [2,34].

The catalysts were tested by cyclic voltammetry (CV) at rates of 100 mV s^−1^ in the potential range from −0.3 to 0.2 V (vs. RHE). The measurements data are plotted as current vs. voltage, also known as a voltammogram exhibited in Figure 4d. A shape with one pair of redox peaks of the CV curve can be observed in the case of the prepared photocatalysts. This obtained phenomenon is due to the surface redox reactions. The peak current for the AgZnS-G-T electrode is higher compared to AgZnS-G and AgZnS. The evaluated amount of the current density of the AgZnS-G-T (7.67 × 10^−4^ mA cm^−2^) is higher than AgZnS-G (4.42 × 10^−4^ mA cm^−2^), and AgZnS (3.22 × 10^−5^ mA cm^−2^) suggest that the AgZnS-G-T photocatalyst can transfer more electrons from VB to CV. From the CV curves, a large current, rectangular-type CV, and symmetric anodic and cathodic scan profile are signatures of the ideal electrical response of photocatalysts for hydrogen production [31]. It is observed that among the three materials in Figure 4d, the CV curves of pristine AgZnS and AgZnS-G display a smaller area than that of the AgZnS-G-T composites. The AgZnS-G-T photocatalyst has the highest electroactivity and largest rectangular areas of the CV curve, indicating the higher ion transport rate and better electrochemical activities of the AgZnS-G-T. This result suggests that introducing RGO and TiO_2_ can improve the electrochemical activity of the AgZnS-G-T composites. Graphene is well known for its large specific surface area, strong electrical conductivity, and high charge carrier mobility. The graphene sheet acts as a support material between TiO_2_ and TMS particles, which may provide a path for the photo-generated electron. Graphene is proposed as a sink for the photo-generated electrons from TiO_2_ or TMS. As a good solid-state electron mediator, the combination of TiO_2_ or TMS can prompt the rapid transfer of photo-generated electrons from the inorganic semiconductor to graphene, reducing the electron-hole recombination rate, which is necessary for achieving higher efficient photocatalytic hydrogen production. The result of the CV test proves that the AgZnS-G-T electrode has excellent electrocatalytic performance for HER [28,31,34].

Electrochemical impedance spectroscopy (EIS) is presented in Figure 5b. The arc radii of the EIS Nyquist curve of the samples were in the following order: AgZnS > AgZnS-G > AgZnS-G-T. AgZnS-G’s arc radius was smaller than AgZnS, and the arc radius of AgZnS-G-T was the smallest. In general, a smaller radius indicated lower charge transfer resistance and higher carrier separation efficiency [35]. That is to say; it has a much faster electron transfer process [36]. Therefore, the EIS test verified that the proper loading of AgZnS and TiO_2_ promoted photoinduced carrier migration between AgZnS, TiO_2_ and RGO and eventually led to a higher photocatalytic hydrogen evolution activity. This was consistent with the results of the photocurrent response.

The linear sweep voltammetry (LSV) curves as shown in Figure 5c. Linear sweep voltammetry characterization was applied to analyze and compare the reduction ability of the AgZnS, AgZnS-G, and AgZnS-G-T composites. Obviously, the AgZnS-G-T shows the lowest overpotential, indicating the highest electrocatalytic activity for HER. To achieve the same current density, the overpotential required by AgZnS-G-T was much smaller than that required by AgZnS and AgZnS-G. In other words, the AgZnS-G-T composite decreased faster than AgZnS and AgZnS-G in the negative direction. These results indicated the presence of charge transfer channels in the AgZnS-G-T compound, which made the excitation and migration of photogenerated electrons easier. The outstanding HER activity of AgZnS-G-T can be further verified by the Tafel plot as displayed in Figure 5d. Tafel plots constructed from LSV data were used for the evaluation of kinetics of HER process. The AgZnS-G-T composite shows the smallest Tafel slope of 22.6 mV·dec^−1^ when compared with 27.5 mV·dec^−1^ for AgZnS-G and 36.4 mV·dec^−1^ for AgZnS. In contrast, the Tafel slope of the commercial Pt/C is 38 mV·dec^−1^.

The photocurrent reflects the density of photoexcited electron-hole pairs under visible light irradiation. Figure 5a shows the photocurrent densities of AgZnS-G-T, AgZnS-G, and AgZnS nanocomposite under visible illumination. The photocurrent density of AgZnS-G-T samples was significantly higher than that of AgZnS-G and AgZnS. The highest photocurrent was obtained by AgZnS-G-T (about 11,906 mA cm^−2^), which was over 1.6 and 2.2 times those of AgZnS-G nanocomposite (about 7400 mA cm^−2^) and AgZnS (about 5352 mA cm^−2^), respectively. The introduction of RGO and TiO_2_ help create a carrier transfer channel to form. The higher photocurrent density of AgZnS-G-T indicated improved optical absorption capability, enhanced separation of photo-generated electron-hole pair, and low recombination rate of the photo-generated electron-hole pair compared to AgZnS and AgZnS-G. With continuous on and/or off cycles of irradiation, the photocurrent density of all samples showed a sharp rise and/or drop. The on–off cycle of the photocurrent was repeatable, indicating that these prepared catalysts had stable photoinduced carrier separation characteristics [31,35]. Photocurrent signal is increased after adhered AgZnS and TiO_2_ to RGO. The superior photocurrent density was attributed to the interfacial charge transfer from the conduction band of TiO_2_ to AgZnS and then transferred to RGO sheets. It was conducted that a helpful modified material was fabricated successfully, which facilitated the rapid transfer and separation of photoinduced carriers.

The semiconductor type and flat-band (Vfb) potential of typical materials were characterized by the Mott–Schottky (M−S) plot at a frequency of 500 Hz. AgZnS, AgZnS-G, and AgZnS-G-T exhibit the positive slope, respectively, which implies that they are n-type semiconducting materials and that the electrons are the main charge carrier. The positive shift in flat band potential may be attributed to two different surface states of materials, which could lead to considerable changes in the band positions. The surface trapped holes of AgZnS, AgZnS-G, and AgZnS-G-T samples occurring electrode/electrolyte interface charge transfer are causing an upward shift in the Fermi level by increasing the degree of band bending and enhancing the efficiency of photo-generated charge carrier’s separation and migration. The energy band structures of pure AgZnS, AgZnS-G, and AgZnS-G-T are investigated by Mott–Schottky plots and UV–vis DRS. The result of Figure 6a expressed that the flat band potentials are −0.450, −0.420, −0.21 V versus standard hydrogen electrode (SHE), respectively. According to the x-intercepts of the linear region, we obtained the flat band potential (Vfb) values of AgZnS, AgZnS-G, and AgZnS-G-T, which is widely thought to be about equivalent to the eCB of n-type semiconductors (CB = Ufb − 0.3 V) [36] due to the conduction band potential (eCB) being more negative by 0.3 V than that of Ufb. Therefore, the conduction band potential (eCB) of AgZnS, AgZnS-G, and AgZnS-G-T were estimated to be −0.534, −0.504, and −0.364 V versus standard hydrogen electrode (SHE, SHE = SCE + 0.413 V), which is slightly negative compared to the water reduction potential. The VB level was obtained by subtracting the Eg energy from the eCB level. The VB positions of pure AgZnS, AgZnS-G, and AgZnS-G-T were calculated as 1.866, 1.746, and 1.236 eV vs. SHE, respectively [31,37]. The band alignment of the as-fabricated photocatalysts was schematically shown in Figure 6b.

### 3.2. Photocatalytic H_2_ Evolution

Figure 7a shows the hydrogen evolution efficiency test of the prepared photocatalysts over one h of equal intervals. The rate of hydrogen evolution continuously increased until the test ended after 4 h. AgZnS-G-T exhibits the highest hydrogen evolution rate. The hydrogen evolution rate of pure TiO_2_ and RGO is not apparent. Furthermore, the HER activities of the several materials were also studied under alkaline, neutral and acidic conditions. Figure 7b showed hydrogen evolution in all pH ranges. All nanomaterials were more active in acidic than in neutrality and alkaline solutions. After 4 h, the AgZnS-G-T electrode had a high hydrogen evolution amount of 930.46 micromole/gat buffer solution (pH = 5), which was superior to other electrodes such as AgZnS-G (790.1 µmole/g) and AgZnS (701.2 µmole/g). It is of great significance to develop stable AgZnS-G-T catalysts at pH = 5 acidic conditions. It was observed that the H_2_ evolution amounts increased with the increased temperature. Figure 7c shows that the hydrogen evolution amount of AgZnS-G-T is the highest at 30 °C reach up to 961 µmole/g. The comparison of hydrogen evolution amount with reference is shown in Table 1. Without the use of sacrificing reagents, the value of prepared photocatalysts is superior to most of the ternary photocatalyst reported before and the commercial Pt catalyst.

Oh et al. carried out H_2_ evolution amounts under ultrasound and light irradiation (sonophotocatalysis) and only under light irradiation [28]. The addition of the ultrasound wave increased the hydrogen evolution amount, which was 4.5, 4.2, and 3.5 times better than that in the absence of ultrasound waves for the BaCuZnS, BaCuZnS-graphene, and BaCuZnS-graphene-TiO_2_ photocatalysts, respectively [28]. The production amount of LaCdSe-GO-TiO_2_ photocatalyst was 397.21 μmol/g for 9 h without implementing ultrasonic oxidation effect. However, the addition of ultrasound has increased the result to 324.15 μmol/g for 1 h [31].

Ultrasonic intensity is one of the important factors affecting H_2_ evolution amounts. The experiments were performed to explore the effects of ultrasonic irradiation power (20, 40, 60, and 80 MHz) on hydrogen evolution. H_2_ evolution amounts have little difference at ultrasonic irradiation power of 20–60 MHz in Figure 7d. The AgZnS-G-T retained the highest H_2_ evolution amounts at ultrasonic irradiation power of 20–60 MHz compared with AgZnS-G and AgZnS. The increase in ultrasonic power promotes the formation of more cavitation bubbles and makes the blasting of cavitation bubbles more intense. H_2_ evolution amounts increased significantly at ultrasonic irradiation power of 80 MHz. AgZnS-G-T demonstrates the higher H_2_ evolution amounts of 985 µmole/g than AgZnS-G (823 µmole/g) and AgZnS (726 µmole/g) at ultrasonic irradiation power of 80 MHz.
nanomaterials-12-03639-t001_Table 1Table 1H_2_ evolution amounts of other catalysts have been reported.CatalystH_2_ Evolution Amounts (with the Scavenger)H_2_ Evolution AmountsReferencesLaCdSe-GO-TiO_2_443.28 μmol/g.324.15 μmol/g/4 h (sonophotocatalytic)[32]BaCuZnS-graphene-TiO_2_5541.04 μmol/g/4 h2715.60 μmol/g/4 h [28]ZnS:Eu quantum dots11100 μmol/g/h9000 μmol/g/h[33]ZnS:V1140 μmol/g/h after 300 min-[38]pure ZnS67 μmol/g/h after 300 min-[13]NiMoS_x_-4.93 μmol/g/1 min[25]CoMoS_x_-3.85 μmol/g/1 min[13]FeMoS_x_-1.57 μmol/g/1 min[13]MnMoS_x_-1.52 μmol/g/1 min[14]platinum-0.35 μmol/g/1 min[14]Ni_6_(SCH_2_Ph)_12_-TiO_2_-5600 μmol/g/h[35]WSe_2_-graphene-TiO_2_2.004 mmol/11 h1.718 mmol/11 h[39]


The reason for the high hydrogen evolution amounts can be summarized as follows. Strong penetration is the characteristic of ultrasonic; the penetration of water medium can reach 15~20 cm. The microwave field can increase the light absorption of the catalyst. Ultrasonic radiations cause acoustic cavitations due to which the formation, growth, and implosive collapse of the bubbles in the liquid phase occur. The acoustic cavitation of ultrasonic in liquid provides high localized pressures and temperatures in effect, the development, growth, and collapse of bubbles. The implosive collapse of the bubbles generates a localized hot spot of extremely high temperature (~5000 K) and high pressure. Due to this high energy and temperature, the nanoparticles accelerate this process is called cavitations. This process has several advantages over other methods of synthesis, including increased mass transfer, a nonhazardous rapid reaction rate, shorter reaction cycles, the production of small nanoparticles, and the homogeneity of the generated nanomaterial [38,39]. The sonoluminescence phenomenon generated by the cavitation effect can produce a wide range of light and heat, part of which can directly excite the semiconductor material so that the electrons in the valence band can be excited to transition to the conduction band, and at the same time leave corresponding holes in the valence band, resulting in electron-hole pair. Inside the prepared nanoparticles, free electrons inside the photocatalyst migrate to up surface along the polarized direction, reducing H_2_O to H_2_. In the meantime, free holes move towards the lower surface and participate in the reaction of OH· generation on the down surface. Ultrasound irradiation exerts a mechanical force on our catalysts, resulting in the separation of interior positive and negative electric charge centers and thus forming an electric field along with the polarized orientation. Consequently, the driving force for carrier separation and migration can be preserved under ultrasonic vibration, enabling a continuous redox reaction [40,41,42]. The role of a high-power sonic wave is to create an electric field in the solution. When the light irradiates on the solution, the created electric field will work as a bridge, which will help transfer electrons from the valence band to the conduction band. The high-power sonic wave will produce superoxide radicals and free hydroxyl radicals (OH·). It will play a vital role in producing photo-generated hole–electron recombination rate and increase the hydrogen evolution amount [29]. Elctrosonophotocatalytic hydrogen evolution mechanism is shown in Figure 8.

Figure 9 showed that the AgZnS-G-T photocatalyst had outstanding long-term stability for 16 h. The H_2_ evolution of the cyclic test of AgZnS-G-T photocatalyst after 4 cycles did not show obvious performance degradation. Thus, the prepared photocatalyst synthesized by the ultrasonic process has good crystalline structural integrity and can be retained after long-time illumination.

## 4. Conclusions

In this work, integrated fabrication of nanosize AgZnS-G-T was synthesized via a soft ultrasonic method. TMS of AgZnS is a new structure. During ultrasonication, simultaneous reduction of GO into RGO and attachment of AgZnS nanoparticles and TiO_2_ are observed in an aqueous solution. Results from various characterized methods demonstrated the formation of well-crystallized photocatalysts with the ultrasonic method. Ag-doped into the ZnS give rise to HER activity enhancement, which can be related to chemical surface structure changes by leading to new bimetallic active sites, including sulfur bridging. The successful combination between TMS and carbon material (graphene) can strengthen the physical and chemical properties of the photocatalysts. The AgZnS-G-T shows higher H_2_ evolution amounts than AgZnS-G and AgZnS. The maximum hydrogen evolution yield of AgZnS-G-T is 930.46 µmole/g at pH = 5. The maximum yield of hydrogen evolution reached up to 961 µmole/g at 30 °C. AgZnS-G-T demonstrates the highest H_2_ evolution amounts of 985 µmole/g at ultrasonic irradiation power of 80 MHz. Sacrificial agent-free photocatalytic hydrogen evolution is environmentally friendly. Therefore, AgZnS-G-T nanoparticles prepared using ultrasonic synthesis can be considered a promising hydrogen evolution catalyst.

## Data Availability

The data can be made available on the basis of request.

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
