# Peer review of "Novel Approach to Synthesis of AgZnS and TiO2 Decorated on Reduced Graphene Oxide Ternary Nanocomposite for Hydrogen Evolution Effect of Enhanced Synergetic Factors"

_nanomaterials, 2022, doi:10.3390/nano12203639_

Round 1

Reviewer 1 Report

The present manuscript discusses on the synthesis and performance evaluation of AgZnS-G-TiOfor photocatalytic HER. Authors have used ultrasonic method for the synthesis and have used several other characterization techniques to support the material developed in the present study. However, the interpretation and discussion are not well represented and I would recommend the authors consider the below comments as resubmit the article for the possible publication. I recommend the authors for the major revision. 

Comments:

1. Authors have calcined at 600 oC to obtain the final material prior to characterization. I suspect the presence of sulfur after calcination at such high temperature. 

2. As discussed by the authors, in XRD both JCPDS are the same and corresponds to the presence of anatase TiO2 and the other diffraction peaks reference is missing. I could suggest the authors to provide reference for ZnS and AgZnS.

3. XPS deconvolution is not acceptable and I recommend authors to repeat and discuss accordingly. 

4. I strongly suggest the authors to use two electrode configuration during catalyst performance evaluation. 

5. Several typos are present and it should be taken care. 

6. Ag/AgCl2 in reference electrode is mentioned. Please correct the typo. 

Author Response

Reviewer 1

The present manuscript discusses on the synthesis and performance evaluation of AgZnS-G-TiO2 for photocatalytic HER. Authors have used ultrasonic method for the synthesis and have used several other characterization techniques to support the material developed in the present study. However, the interpretation and discussion are not well represented and I would recommend the authors consider the below comments as resubmit the article for the possible publication. I recommend the authors for the major revision.

Comments:

  1. Authors have calcined at 600 ℃ to obtain the final material prior to characterization. I suspect the presence of sulfur after calcination at such high temperature.

Response: Thank you for your comment. We suggest the evidences which the presence of sulfur was verified by EDS、XRD、XPS and other technical means.

  1. As discussed by the authors, in XRD both JCPDS are the same and corresponds to the presence of anatase TiO2 and the other diffraction peaks reference is missing. I could suggest the authors to provide reference for ZnS and AgZnS.

ResponseXRD pattern for AgZnS shows no extra peaks or change in the position of present peaks compared with ZnS nanoparticles. The XRD patterns of prepared samples revealing the structure of the ZnS were not influenced by the Ag ion. This observation suggests that AgZnS nanoparticles are formed in a single-phase and Ag+ ions have incorporated into the ZnS hexagonal lattice without forming any different compound. There are silver and zinc characteristic peak in the EDS. We already suggested AgZnS reference on XRD patterns. But, we can find ZnS combined nanocomposite with ZnS trace peaks on reference. No single phase of ZnS was found from the XRD pattern. The study of the combined complex of ZnS can be seen from our previously published work [27]. Journal of Ceramic Processing Research. Vol. 13, No. 3, pp. 283~290 (2012)

  1. XPS deconvolution is not acceptable and I recommend authors to repeat and discuss accordingly.

Response: Thanks for the advice. We additionally described for your comments with paper citation. Journal of Materials Science: Materials in Electronics, 2021, 32:9804-9821. Accurate chemical analysis of graphene-based materials using X-ray photoelectron spectroscopy. Carbon, 2019, 143: 268-275. The peaks located at (368.1 and 374.1) eV in Fig. 3(b) are ascribed to Ag 3d. It is characteristic peak of zero-valent Ag. The peak of oxygenated carbon in RGO decreased significantly even disappeared, indicating that GO was gradually reduced to RGO in the preparation process.

  1. I strongly suggest the authors to use two electrode configuration during catalyst performance evaluation.

Response: Thanks for the advice. Experimental data is being supplemented. The manufacturing process and electrochemical experimental method of the sample are presented in Scheme 1.

  1. Several typos are present and it should be taken care.

Response:Thanks for the advice. The misspelling has been corrected.

  1. Ag/AgCl2 in reference electrode is mentioned. Please correct the typo.

Response: Thanks for the advice. This typo has been corrected.

Reviewer 2 Report

This manuscript offers a concise yet compelling report on the synthesis of reduced graphene oxide decorated with AgZnS and TiO2 which represents a ternary nanocomposite with a claimed application for hydrogen production. The emphasis of the study falls on the achieved result revealing that AgZnS-TiO2–RGO has the highest efficiency of photocatalytic activity for hydrogen production by water splitting. The discussion provided is quite adequate for the present ambitious purpose. The nice detailed and at the same time comparative context of the synthesis procedure and especially of the detailed and multifaceted characterization results throughout the study contributes to a reliable scrutinization and prompting good understanding of possibly for reliable and repeatable realization of the AgZnS-TiO2–RGO as well as paving the way to its future applications for hydrogen production with potentially high practical impact.

From practical point of view, the reported results thus bring new knowledge and certainly represent an original contribution in the present context.

The authors chose an adequate structure of the manuscript – an excellent point of departure for such a study. Finally, the authors provided a balanced realistic and nicely illustrated presentation of their results and corresponding analysis that is of much scientific and practical interest and adds new knowledge to the field.

In my opinion, the fine detailing in the present work, the insightful and balanced discussion of the results, as well as the excellent, intuitively perceived figures, permit wide circle of readers to utilize the manuscript as a guidance for their potential future work in the same or in a similar research field. Consequently, this manuscript presents an efficient and beneficial basis for promoting and solving next step challenges in this field.

The manuscript also benefits from a clear motivation, and it is an easy and informative read.

The present manuscript is a significant contribution, this work once published would be quite useful as well as instructive and suggestive in terms of further studies and to a wider readership.

There are some minor issues with this already excellent manuscript that will need to be addressed before becoming suitable for publication, i.e., it can be considered for publication after a minor revision:

1: Title is not optimal; it is difficult to understand and grammatically incorrect. Title should clearly state: “Novel approach to synthesis of AgZnS-and-TiO2–decorated reduced graphene oxide with applications for hydrogen production”. No abbreviations as RGO in the title are advisable. In addition, it is grammatically incorrect (including in titles) to begin every word in a sentence or a phrase with capital letters (although some journals accept such a practice).

2: In the introduction, the authors miss that previously a very wide range of theoretical/simulation approaches/including by using first-principles calculation tools have already been used to study decoration and structure growth together in complex, 2D-like, and/or nanostructured materials. Examples in which such theoretical works help understanding such synergies and structural issues and directly guide experimental work include Journal of Physics: Condensed Matter 27 (2015) 485306, ACS applied materials & interfaces 10 (2018) 16238-16243; Such works should be referred to.

3: The authors should elaborate and be more specific when they comment on the thermal stability of the decorated samples in the context of different temperatures throughout the synthesis and characterization. Are there any direct constrains of its thermal stability?

4: It would be helpful and valuable to the general readership if changes in carbon bonding (especially when XPS results are discussed) is commented in more quantitative details and if it is placed in a larger context of bonding in similarly decorated materials.

5: Spell-check and stylistic revision of the paper are still necessary. Some, long sentences, misspellings, etc., still are noticeable throughout the text.

Author Response

Reviewer 2

There are some minor issues with this already excellent manuscript that will need to be addressed before becoming suitable for publication, i.e., it can be considered for publication after a minor revision:

1: Title is not optimal; it is difficult to understand and grammatically incorrect. Title should clearly state: “Novel approach to synthesis of AgZnS-and-TiO2–decorated reduced graphene oxide with applications for hydrogen production”. No abbreviations as RGO in the title are advisable. In addition, it is grammatically incorrect (including in titles) to begin every word in a sentence or a phrase with capital letters (although some journals accept such a practice).

Response: Thank you for your comment. We revised title as you required.

2: In the introduction, the authors miss that previously a very wide range of theoretical/simulation approaches/including by using first-principles calculation tools have already been used to study decoration and structure growth together in complex, 2D-like, and/or nanostructured materials. Examples in which such theoretical works help understanding such synergies and structural issues and directly guide experimental work include Journal of Physics: Condensed Matter 27 (2015) 485306, ACS applied materials & interfaces 10 (2018) 16238-16243; Such works should be referred to.

Response: References are added in the introduction as follow: ” The first-principles calculations based on density functional theory provide a possible understanding of surface interactions between RGO decoration with other transition metal [22,23].”

3: The authors should elaborate and be more specific when they comment on the thermal stability of the decorated samples in the context of different temperatures throughout the synthesis and characterization. Are there any direct constrains of its thermal stability?

Response : Thank you for your comments. We have established synthesis methods for ZnS, TiO2 and AgZnS based on the methods presented below. This temperature was set in consideration of the chemical bonding of the S element. In addition, in the case of TiO2, a phase change occurs at 600 ℃ or higher, so it was set to an appropriate temperature. Fullerenes, Nanotubes and Carbon Nanostructures, Vol. 22, No. 7, 01 May 2014. C. Y. Park, J. G. Choi, T. Ghosh, Z. D. Meng, L. Zhu, W. C. Oh, Preparation of ZnS-Graphene/TiO2 Composites designed for the High Photonic Effect Under Visible Light, Proceeding of Annual World Conference on Carbon, CARBON 2012.

4: It would be helpful and valuable to the general readership if changes in carbon bonding (especially when XPS results are discussed) is commented in more quantitative details and if it is placed in a larger context of bonding in similarly decorated materials.

Response: Thank you for your comments. We additionally described for your comments with paper citation. Journal of Materials Science: Materials in Electronics, 2021, 32:9804-9821. Accurate chemical analysis of graphene-based materials using X-ray photoelectron spectroscopy. Carbon, 2019, 143: 268-275. The peaks located at (368.1 and 374.1) eV in Fig. 3(b) are ascribed to Ag 3d. It is characteristic peak of zero-valent Ag. The peak of oxygenated carbon in RGO decreased significantly even disappeared, indicating that GO was gradually reduced to RGO in the preparation process.

5: Spell-check and stylistic revision of the paper are still necessary. Some, long sentences, misspellings, etc., still are noticeable throughout the text.

Response: Thanks for your suggestion. We apologize for this ambiguity. The misspelling has been corrected.

Round 2

Reviewer 1 Report

Thanks to the authors for the revision. The work describes the photocatalytic HER and it is interesting in the real water electrolyser application. However, XRD patterns are not well explained with JCPDS reference and XPS analysis is not improved. The background in XPS during deconvolution is not acceptable and can be improved so that much more results can be discussed to support AgZnS. In addition, I would strongly recommend the authors to provide HER in two electrode configuration with photocatalytic effect.

Author Response

Thank you for your valuable comments. We revise XRD parts with new plotted pattern and JCPDS marking. And, all of XPS data re-plotted with new deconvolution. Fig. 8 revised with schematic electrodes system.
